# A Randomized Controlled Trial of a Parent-Led Memory-Reframing Intervention to Reduce Distress and Pain Associated with Vaccine Injections in Young Children

**DOI:** 10.3390/children10071099

**Published:** 2023-06-22

**Authors:** Maria Pavlova, Atiqa F. Pirwani, Jody Thomas, Kathryn A. Birnie, Michelle Wan, Christine T. Chambers, Melanie Noel

**Affiliations:** 1Department of Psychology, University of Calgary, Calgary, AB T2N 1N4, Canada; mpavlova@ucalgary.ca (M.P.); atiqa.pirwani@ucalgary.ca (A.F.P.); 2Department of Psychiatry and Behavioral Sciences, Stanford University School of Medicine, Stanford, CA 94305, USA; jody.thomas@megfoundationforpain.org; 3Meg Foundation, Denver, CO 80238, USA; 4Department of Anesthesiology, Perioperative, and Pain Medicine, University of Calgary, Calgary, AB T2N 1N4, Canada; kathryn.birnie@ucalgary.ca; 5Department of Community Health Sciences, University of Calgary, Calgary, AB T2N 1N4, Canada; 6Alberta Children’s Hospital Research Institute, University of Calgary, Calgary, AB T2N 1N4, Canada; 7Hotchkiss Brain Institute, University of Calgary, Calgary, AB T2N 1N4, Canada; 8Owerko Centre, University of Calgary, Calgary, AB T2N 1N4, Canada; 9Mathison Centre for Mental Health Research & Education, University of Calgary, Calgary, AB T2N 1N4, Canada; 10Solutions for Kids in Pain, Halifax, NS B3H 0A8, Canada; 11Department of Psychology & Neuroscience, Dalhousie University, Halifax, NS B3H 4R2, Canada; christine.chambers@dal.ca; 12Department of Pediatrics, Dalhousie University, Halifax, NS B3H 4R2, Canada; 13Centre for Pediatric Pain Research, IWK Health Centre, Halifax, NS B3K 6R8, Canada

**Keywords:** parent–child reminiscing, memory-reframing intervention, needle pain, pediatric pain, memory of pain, COVID-19 vaccine injection

## Abstract

Children remember their memories of pain long after the painful experience is over. Those memories predict higher levels of future pain intensity. Young children’s memories can be reframed to be less distressing. Parents and the way they reminisce about past events with their children play a key role in the formation of pain memories. A novel parent-led memory-reframing intervention changed children’s memories of post-surgical pain to be less distressing. The intervention efficacy in the context of vaccine injections is unclear. This registered randomized controlled trial (NCT05217563) aimed to fill this gap. Seventy-four children aged 4.49 years (*SD* = 1.05) and scheduled to obtain two COVID-19 vaccine injections and one of their parents were randomized to receive: (1) standard care; (2) standard care and memory-reframing information; and (3) standard care and memory-reframing information with verbal instructions. Children reported their pain after vaccine injections. One week after the first vaccination, children reported memory of pain. Parents reported their use of memory-reframing strategies and intervention feasibility and acceptability. The intervention did not result in significant differences in children’s recalled or future pain. Parents rated the intervention as acceptable and feasible.

## 1. Introduction

Children experience pain often and remember it long after the painful event is over. The COVID-19 pandemic brought on an urgent and continuing need for children to receive repeated vaccine injections and boosters, adding further to the Canadian routine childhood immunization schedule that includes 20 vaccine injections before age 5 [1]. Pain associated with medical procedures (e.g., vaccine injections) is considerable with most children experiencing at least moderate levels of pain during vaccine injections [2]. Children’s gender, temperament, and other individual characteristics influence the experience of pain [3]. For example, younger children and those with more active temperament exhibit higher levels of distress during immunizations [3]. These pain experiences, as well as resulting memories of pain, leaves a lasting impact in the form of distress, needle fears, and vaccine hesitancy [4,5].

Children remember their past painful experiences, and those memories predict future levels of pain better than the initial pain experience [1]. This has been demonstrated for experimental, post-surgical, and procedural pain experiences [6]. The importance of memory of the present- and future- moment experiences may be due to our reliance on the same brain structures (i.e., medial temporal lobe) when remembering past events and preparing for future events [7]. In other words, our memories prepare us for the future. Children’s memories of pain are fragile and can become distorted over time. Most children remember their pain accurately or recall less pain than they initially reported (i.e., remembering pain in a positively biased way). For example, a child may rate their pain intensity during a flu vaccination as 5 out ot 10. When asked how much pain the child remembers a week later, the child may recall a 5/10 pain intensity (i.e., accurate recall) or remember less pain intensity (e.g., 2/10; positively biased pain memory). However, up to a quarter of children develop negatively biased memories of pain (i.e., remembering higher levels of pain as compared to their initial experience; e.g., recalling 8/10 pain intensity instead of the initially reported 5/10). Negatively biased pain memories are associated with higher levels of pain intensity during future painful procedures and more distress [8,9]. Children who are more anxious [10] experience greater pain [11] and oftentimes have parents who are more anxious [12] and who tend to catastrophize their children’s pain [13]. Therefore, such children are at a greater risk of developing negatively biased pain memories.

Children’s memories of pain can be changed to be less distressing and/or more accurate. Effective pain management (e.g., the use of topical anesthetics) can protect children from developing negatively biased pain memories [14,15]. Memory-reframing interventions have also proven to be an effective way to change pain memories. Existing memory-reframing interventions have been delivered by researchers, have been conducted for procedural pain (e.g., lumbar punctures, dental injections), and have resulted in children developing less negatively biased memories of procedural pain; however, the interventions did not decrease children’s anticipatory fear or overall distress (for a full review, see [16]). An increased pain-related self-efficacy (i.e., one’s belief that they are able to cope with pain) was proposed as a possible mechanism [17,18]. However, given the pivotal role that parents play in the development and modification of children’s memories [19], including memories of pain [20], it was argued that parents should successfully apply memory-reframing strategies to change their children’s memories of pain [21].

A novel parent-led memory-reframing intervention was developed to utilize the malleability and parental role in the formation of children’s memories. The intervention focused on emphasizing positive aspects of past painful experiences, reframing any overly distressing memories, and scaffolding children’s self-efficacy [22]. The intervention changed children’s memories of post-surgical pain to be more accurate/more positively biased [22]. The intervention focused on increasing children’s self-efficacy related to coping with pain. To achieve this goal, the intervention combined the elements of previously used strategies (e.g., discussing coping strategies employed, highlighting children’s bravery in the context of painful procedures, correcting negative biases in recalling past pain) [17,18,23] and observational data demonstrating the associations between elaborative parental reminiscing style (i.e., using open-ended questions, discussing emotions) and children’s less negatively biased memories of pain [20]. The intervention was feasible and acceptable in the context of minor and major surgeries and changed parental reminiscing style to be more adaptive (i.e., asking open-ended questions, using more emotion-laden and less pain-/fear-related words) [22,24]. Further, the intervention resulted in young children developing more accurate/positively biased memories of pain associated with minor surgery [22].

However, the efficacy of the parent-led intervention to change children’s memories of needle pain, as well as to reduce the levels of future pain, is unknown. Further, it is not clear whether the delivery of the intervention to parents requires an interventionist or whether a summary of the intervention principles in a form of a handout and/or video would be sufficient to produce the beneficial effects. The present study aimed to fill these gaps. Specifically, this randomized controlled trial (RCT) examined the efficacy of a parent-led memory-reframing intervention on changing children’s memories of pain and fear to be more accurate/positively biased and on reducing future pain and fear associated with repeated COVID-19 vaccine injections. Given the previously demonstrated efficacy of the intervention [22], we hypothesized that the intervention would result in children whose parents received intervention to develop more accurate and/or positively biased memories of pain intensity and pain-related fear. As previous observational data showed a link between more negatively biased memories of pain and future pain and fear [11], we hypothesized that the intervention would lead to children reporting less pain and fear during their second COVID vaccine injections. It was also hypothesized that the magnitude of the intervention effect would differ depending on group allocation: children whose parents received verbal instructions in addition to intervention materials and standard care would report more positively biased/accurate memories of pain and fear, as well as lower levels of future pain and fear, compared to children whose parents received only intervention materials and standard care, followed by children receiving standard care only. Alternatively, there would be no statistically significant differences between the groups. We hypothesized that the intervention would be acceptable and feasible. Parents receiving additional verbal instructions from an interventionist would adhere to the intervention principles to a greater extent compared to parents receiving only intervention materials and standard care (and no verbal instructions). Finally, exploratory analyses examined the effects of the intervention on children’s expected pain and fear, parent and child pain-related self-efficacy, as well as the role of individual and therapeutic factors on the effects of the intervention.

## 2. Materials and Methods

### 2.1. Trial Design

This RCT was pre-registered on clinicaltrials.gov (NCT05217563; posted 1 February 2022) and had a parallel group assignment with a 1:1 allocation ratio. Researchers assessing primary outcomes of memory of pain intensity and pain-related fear were blinded to group allocation. Parents who collected primary outcomes of pain intensity and pain-related fear associated with the second COVID vaccine injection were not blinded to group allocation. Participant recruitment occurred from February 2022 to November 2022. Recruitment ended due to the discontinuation of study funding (see Protocol Deviations section below).

The study was advertised on social media (i.e., Twitter, Facebook) and pediatric pain management websites and newsletters (e.g., Solutions for Kids in Pain), as well as promoted among daycares across Canada. Interested families were invited to contact study staff. If eligible, parents completed consent forms and sociodemographic questionnaires using an online survey software [25]. Data collection followed the pre-registered study protocol (Figure 1). Upon providing consent, which was on average 1.5 weeks prior to the first COVID vaccination appointment, all parents received information regarding evidence-based needle pain management strategies (e.g., distraction, topical anesthetics; see Appendix A) [26]. Parents randomized to either of the intervention groups (see Section 2.6) received information summarizing intervention strategies (see Appendix A). Parents assigned to an intervention group with an interventionist additionally received verbal instructions on memory-reframing strategies. Children reported their pain intensity and pain-related fear immediately after the first COVID vaccine injection using validated faces pain and fear scales [27,28]. The ratings were obtained by a parent who participated in the study and was present for the vaccination appointments. Similar to previous research [22,29], parents were instructed on how to administer the scales to their children; parents received the faces scales on the day of the vaccination appointments via the online survey software (i.e., REDCap [25]) and obtained children’s ratings within 10 to 15 min after the vaccine injection. Seven to twelve days (*M* = 8 days, *SD* = 1.28) after the first COVID vaccine injection, children completed an established memory interview [20] via telephone or virtual conference platform (i.e., Zoom). During the interview, children reported their memory of pain intensity and pain-related fear associated with the first COVID vaccine injection. Children also reported how much pain and fear they expected to experience during the second COVID vaccine injection. Trained research assistants who were blind to group allocation obtained ratings for memory of pain and pain-related fear using the same faces scale previously administered. Children received their second COVID vaccine injection on an average of 9 weeks (*SD* = 4.05) after the first vaccination appointment. Parents obtained children’s ratings of pain and pain-related fear within 10 to 15 min after the vaccine injection using the same faces scales. Parents reported their use of pain management strategies. Parents assigned to intervention groups rated intervention acceptability and feasibility and reported on their use of the intervention principles in between vaccination appointments. Parents assigned to the standard care group received the summary of the intervention principles via email upon study completion. As a token of appreciation, participants received two CAD 20 gift cards (i.e., after the memory interview and after the second vaccine appointment).

### 2.2. Protocol Deviations

There was one change to the trial protocol after the study commencement. Specifically, the original trial was registered to include children aged 5 to 11 years. After the Public Health Agency of Canada approved COVID vaccines for children under 5 years on 14 July 2022, and given the efficacy of the memory-reframing interventions for children as young as 4 years [22], the age range was expanded to include children aged 4 years.

The registered study protocol specified recruiting 300 parent–child dyads. The trial funding was secured for one year starting January 2022. Once the ethical approval and trial registration was obtained (i.e., February 2022), the study was advertised, as described above. The recruitment rate was not as high as initially anticipated. Pediatric vaccine uptake was fast with 90% of initial doses administered in Alberta within two months after the vaccine approval [30]. Fast vaccine uptake combined with the reliance on potential participants to contact the research team resulted in slower-than-anticipated recruitment. Additional advertisement and further contact with vaccination clinics did not change recruitment rates. By November 2022, it was evident that the target size (i.e., 300 parent–child dyads) would not be recruited by the end of funding period (i.e., December 2022). Therefore, it was decided to stop the trial before reaching the target size. The trial aims and measures remained the same. The study results should be interpreted in light of the small sample size.

### 2.3. Randomization and Blinding

An independent researcher performed block randomization (1:1) using a random number generator [31] and entered group allocation into a password-protected file not accessible to the rest of the research team (i.e., interventionists, fidelity coders, memory interviewers). Research staff that was not involved in the study had access to group allocation information and assigned interventionists based on their availability to meet with parents randomized to receive verbal instructions. Apart from delivering the verbal instructions, interventionists did not interact with participants. Participants’ group allocation was concealed from the researchers assessing the primary outcomes of their memories of pain and pain-related fear. Group allocation could not be concealed from parents who collected the primary outcome of pain intensity and fear after the second COVID vaccine injection. The first author (MP) performed statistical analyses after data collection was completed; the designation of group allocation was concealed (i.e., groups were labeled as A, B, and C).

### 2.4. Participants

Out of 148 parent–child dyads screened for eligibility, 74 children aged 4–11 years and one of their parents living in Canada participated in the study. Children were eligible for the study if they were between 4 and 11 years of age, scheduled to receive a two-dose COVID vaccine, able to understand and speak English, and had a caregiver willing to participate in the study and able to understand and speak English. Exclusion criteria included developmental disabilities (e.g., autism spectrum disorder) and/or language delays as these conditions would make it challenging for participants to complete the study tasks and measures.

Similar to previous research [20,22], the age range of 4 to 11 years was chosen due to children’s rapidly developing language and cognitive skills. Additionally, preschool-aged children begin to actively participate in reminiscing about past events with their parents, who play a key role in developing children’s autobiographical memories through reminiscing [32]. Used during preschool and early school years, memory-reframing interventions harness both the malleability of children’s autobiographical memories and parents’ notable influence on children’s memories.

### 2.5. Ethics

All participants received information regarding evidence-based needle-related pain management strategies [33]. The institution’s research ethics board approved the study (REB21-1997). No adverse events or unintended effects were reported.

### 2.6. Interventions

#### 2.6.1. Standard Care Group

Parents randomized to the standard care group received information on evidence-based strategies to reduce pain and fear associated with vaccine injections (e.g., distraction, topical anesthetics) [26]. Parents in the standard care group did not receive any information regarding memory reframing, nor were they encouraged to discuss vaccine injection experience with their children. Parents in the standard care group received text/email reminders to use needle pain management strategies before their children’s first and second vaccination appointments.

#### 2.6.2. Intervention Groups

Parents assigned to either of the two intervention groups received the same information regarding evidence-based strategies to manage needle pain and distress as described above, as well as text reminders to use the pain management strategies before vaccination appointments. Parents in both intervention groups also received a pamphlet summarizing memory-reframing strategies to be used when discussing vaccination appointments (Appendix A). The pamphlet also contained a link to a video [34] that explained the intervention principles (the video is available for public consumption for patient education). Intervention group 1 (information only) received only the pamphlet. Intervention group 2 (information + verbal instructions) received the pamphlet as well as verbal instructions on how to use these principles. Verbal instructions were delivered by trained clinical psychology graduate students and the senior author, a registered psychologist, via telephone or virtual conference platform. Specifically, the interventionist reviewed the memory-reframing strategies with parents assigned to intervention group 2 and elicited specific examples of their child’s bravery, successful coping with pain, and positive aspects that can take place during or after vaccine injection. Parents in both intervention groups received text/email reminders to discuss children’s COVID vaccination experience using the memory-reframing principles; the reminders were sent approximately once every two weeks starting after the first and until the second COVID vaccination appointment. The intervention instructions lasted, on average, 16 min (*SD* = 6.39); most parents requested the intervention instructions to be delivered via video conferencing (i.e., Zoom; 81%).

The intervention was based on the parent-led pain memory-reframing intervention that changed pain memories of young children undergoing tonsillectomies to be more accurate or more positively biased [22]. In line with the existing intervention, parents in this study were taught to reminisce about past painful experiences in optimal ways by: (1) focusing on any positive aspects that happened during or after the vaccination appointment and avoiding the use of fear- and pain-related words (e.g., “scared”, “painful”) [20]; (2) correcting any exaggerations with regard to memories of the vaccine injection; and (3) enhancing children’s pain-related self-efficacy by praising their bravery, giving examples of how children were brave, and discussing helpful coping strategies that the children had previously used at the first vaccine injection [18]. The intervention was modified to accommodate parental beliefs regarding reminiscing about past pain [35]. Specifically, to address parents’ tendency to avoid reminiscing about pain as it may be frightening and/or distressing for children, the interventionists highlighted the benefits of an optimal reminiscence of pain and addressed any expressed parental ambivalence to discuss past pain with their children. Additionally, given the potential of reminiscing to complement preparatory interventions for painful procedures [36], parents were encouraged to bolster preparation for the second vaccination by reminding children what helped them during the first vaccination and expressing their belief that the second vaccination appointment would progress well. Interventionists emphasized that not lying regarding children’s experiences was essential. Specifically, if a child was truly distressed during the vaccine injection and recalls the distress, parents were encouraged to acknowledge the distress and help children reframe it by using the strategies described above. For parents allocated to the intervention group 2, interventionists provided suggestions for specific questions and remarks which parents could make while reminiscing about the first vaccination appointment.

#### 2.6.3. Intervention Fidelity

The first and senior authors (MP, MN) developed the intervention summary and a standardized script to follow when delivering the intervention instructions. The first author trained clinical psychology graduate students to deliver the intervention using the script. Intervention delivery was audio-recorded. The intervention delivery was coded for intervention fidelity using a previously used checklist [22]. Two researchers not involved in the study coded the intervention instructions with inter-coder reliability > 0.99 (Cohen’s kappa).

### 2.7. Outcomes

#### 2.7.1. Primary Outcomes

Children’s memory biases for pain intensity and children’s memory biases for pain-related fear during the first COVID vaccination injection were examined. In line with previous research [20,22], memory biases were defined as within-person deviations between the initial report of and recalled pain intensity and pain-related fear. Lower levels of recalled pain intensity or pain-related fear, compared to the initial ratings, represented positively biased pain memories (i.e., remembering less pain/fear compared to the initial reports). Higher levels of recalled pain intensity or pain-related fear, compared to the initial ratings, represented negatively biased pain memories (i.e., recalling more pain/fear compared to the initial reports).

Children reported their memory of pain and fear associated with their first COVID vaccination injection, on average, 8 days (*SD* = 1.28, range 7 to 12) after the vaccination appointment. Trained researchers blind to group allocation obtained the ratings. Faces pain and fear scales (described below) [27,28] used to assess initial reports of pain and fear were administered during the memory interview [22].

Children’s pain intensity and pain-related fear during the second COVID vaccine injection were examined. Both pain intensity and pain-related fear were assessed to capture the multidimensional pain experience (i.e., sensory and affective aspects of pain) [20,37]. Pain intensity, as well as memory of pain intensity, were assessed using the faces pain scale—revised (FPS-R) [27]. The FPS-R consists of six faces that depict pain expressions ranging from neutral (0; no pain) to extreme pain (10; worst pain possible). Children’s pain-related fear, as well as memory of pain-related fear, were assessed using the children’s fear scale (CFS). The CFS depicts five faces that range in their expression of fear from no (0; not at all scared) to extreme (4; most scared possible) fear. The FPS-R and the CFS have been previously used to assess pain intensity, pain-related fear, and memory of pain and fear in children aged 4 to 12 years [11,22]. Both scales are reliable and valid [27,28].

#### 2.7.2. Secondary Outcomes

Child and parent self-efficacy: Parents and children rated their beliefs regarding their ability to cope with their own (children), or to help coping with children’s (parents), needle pain during the second COVID vaccine injection (i.e., pain-related self-efficacy). Children used a numerical rating scale (NRS) ranging from 0 (not well at all) to 10 (very well) to rate “How well do you think you will do when you get your second COVID vaccine shot” [18]; parents used the same NRS to answer “How well do you think you will be able to help your child at their second COVID vaccine shot?”.

Feasibility and acceptability: In line with previous research [22,38], the proportion of enrolled-to-recruited participants was used to assess feasibility. Parents assigned to the intervention groups complemented the feasibility assessment by rating their motivation to learn, their understanding of the intervention, and parent–interventionist rapport (intervention group 2 only) using 0–10 Likert scales; higher scores represent higher levels of feasibility. Further, parents allocated to either of the intervention groups rated intervention acceptability using a 9-item reliable and valid treatment evaluation inventory-short form (TEI-SF) [39]; internal consistency in the current sample was good (Cronbach’s α = 0.81).

Adherence: Parents reported their use of recommended pain management (all groups) and pain memory-reframing strategies (intervention groups) using a parent adherence checklist (Appendix A).

Expected pain and pain-related fear: At the end of the memory interview, children rated how much pain and pain-related fear they expect to experience during their second COVID vaccine injection using the FPS-R and the CFS (described above).

#### 2.7.3. Baseline and Additional Measures

Sociodemographics and medical history: Parents reported their and their children’s sociodemographic information (i.e., age, gender, race/ethnicity), their own level of education, marital status, and household income. Parents also reported their children’s brief medical history.

Parent trait anxiety: Parents used the trait subscale of the state-trait anxiety inventory (STAIT), a 40-item self-report measure of state and trait anxiety with excellent psychometric properties of reliability and validity [40], to rate their trait anxiety [41]. Higher scores obtained using 4-point Likert scales represent higher levels of anxiety. The STAIT’s internal consistency was excellent (Cronbach’s α = 0.93) in our sample.

Parent catastrophic thinking about child pain: Parents’ tendency to magnify, ruminate about, and feel helpless regarding their children’s pain was rated using the pain catastrophizing scale-parent version (PCS-P). The 13 PCS-P items are rated on a 5-point Likert scale with higher scores representing higher levels of catastrophic thinking about child pain. The PCS-P has demonstrated good validity and reliability in parents of youth with and without pain problems [42]. In our sample, the measure’s internal consistency was excellent (Cronbach’s α = 0.92).

Child trait anxiety: Parents reported their children’s anxiety using the PROMIS parent proxy short form-anxiety (ver. 2), a reliable 8-item measure of cognitive, somatic, and affective anxiety symptoms [43]. The items are rated on a 5-point Likert scale and converted to t-scores with higher scores representing higher levels of anxiety.

Needle fears: Parents reported their children’s and their own needle fear using an 11-point NRS (0, not at all scared, to 10, most scared possible).

Vaccine hesitancy: Parental vaccine hesitancy was measured using the 15-item parent attitudes about childhood vaccines (PACV), a valid and reliable measure assessing parent beliefs, behaviours, attitudes, and trust regarding pediatric vaccination [44]. Higher scores represent higher levels of vaccine hesitancy.

### 2.8. Sample Size and Power

GPower (v.3.1.9.2) software was used to conduct an a priori power calculation [45]. The primary outcome of children’s memories of pain was chosen to conduct an ANCOVA family of tests. Previous parent-led, as well as researcher-delivered, memory-reframing interventions demonstrated a medium effect on children’s memories of pain (*η_p_*^2^ = 0.07) [16,22]. To detect a medium effect size with a 0.05 type I error, and 80% probability, a sample of 158 children was required. The registered sample was inflated to *N* = 300 to account for attrition. The study was discontinued due to challenges with meeting the target sample size within the funding period; at the time of study discontinuation, 77 dyads had been randomized.

### 2.9. Statistical Methods

Data analyses were conducted using SPSS (v.29) [46]. The sample, as well as intervention acceptability and feasibility, was characterized using descriptive statistics. Sociodemographic variables were compared across groups using one-way analyses of variance (ANOVA) and chi-squared and Fisher exact tests. Any statistical differences detected by ANOVAs were followed up by independent sample *t*-tests with Bonferroni correction.

In line with previous research [22] and to account for strong associations between initial pain/fear reports and memory of pain/fear [11], children’s memories of pain and pain-related fear were modelled using one-way analysis of covariance (ANCOVA) models. The categorical variable was group allocation (i.e., standard care group, intervention group 1, or intervention group 2) with initial reports of pain/pain-related fear included as covariates (e.g., initial reports of pain intensity associated with the first COVID vaccine injection were covaried in the model assessing memory of pain intensity) [20,22]. Previous research demonstrated meaningful differences during future painful procedures between children who remembered more pain compared to initial reports, versus children who recalled their pain accurately or underestimated their past pain [11]. Therefore, accurate and positively biased (i.e., recalling less pain than reported initially) memories of pain were combined to compare with negatively biased (i.e., recalling more pain than reported initially) pain memories.

Pain intensity and pain-related fear ratings during the second COVID vaccine injection were analyzed using ANCOVAs with the categorical variable of group allocation, while controlling for children’s initial pain and fear ratings obtained immediately after the first COVID vaccine injection. A complete case intent-to-treat approach was used to analyze both sets of primary outcomes. The effect of the intervention on children’s anticipatory pain and pain-related fear was examined similarly using a series of ANCOVA models. Significant models were followed up with independent samples *t*-tests with the Benjamin–Hochberg-corrected alphas.

Exploratory analyses were conducted to investigate the role of individual and therapeutic characteristics on intervention efficacy. Within-subject ANCOVAs were conducted to examine the changes in parent and child pain-related self-efficacy as a function of group allocation. Using partial correlations to control for the initial ratings of pain and fear, we examined associations between children’s (i.e., age, gender, ethnicity, needle fear, anxiety, pain-related self-efficacy) and parents’ (i.e., gender, income, education, ethnicity, anxiety, needle fear, catastrophic thinking about children’s pain, pain-related self-efficacy) individual characteristics and children’s memory of pain and fear, as well as children’s reports of pain and fear associated with their second COVID vaccine injection. Similarly, parent-reported factors associated with the intervention delivery (i.e., motivation to learn and understanding of the motivation, parent–interventionist rapport) and parent-reported use of memory-reframing strategies were partially correlated with children’s memories of pain and fear, as well as their pain and fear levels during the second vaccine injection, while controlling for the initial reports pain and fear. Only children–parent dyads who were allocated to, and who received, the intervention were included in these analyses (*n* = 39); only parents who received verbal instructions (i.e., intervention group 2) were included in the analyses investigating the role of parent–interventionist rapport (*n* = 16). Given the exploratory nature of these analyses, Bonferroni-corrected alphas (α = 0.005) were used to control for type I error.

## 3. Results

The recruitment was conducted from February 2022 to November 2022, and the study was concluded by January 2023. Seventy-seven parent–child dyads were randomized (Figure 2).

Three parent–child dyads were lost prior to completing baseline questionnaires (i.e., one dyad obtained a vaccine without informing the study team; two dyads stopped answering emails). Two families withdrew from the study prior to the first vaccination appointment (i.e., stopped returning emails). One parent–child dyad did not complete the memory interview as the child felt too shy to talk on the phone; the family did complete pain and fear ratings for the second vaccination injection. Six families were lost to follow-up prior to the second vaccination appointment (i.e., one family decided against receiving the second dose of COVID vaccine; five families stopped returning emails). Participants who completed the study did not significantly differ from those who withdrew from the study on key sociodemographic variables. Baseline data from 74 dyads were analyzed. For the primary outcome of memory of pain/fear, data from 71 dyads were analyzed. For the primary outcome of pain associated with the second vaccine injection, data from 65 dyads were analyzed; for the primary outcome of fear associated with the second injection, data from 66 dyads were analyzed. Data were assumed missing at random (little MCAR test *p* = 0.86). No data imputations were performed.

The sample (89% mothers, 57% girls, child *M_age_* = 4.5 years) was predominantly white, educated (89% of parents held a college degree), and had income above CAD 70,000 (88% of parents) (Table 1). Detailed information on participants’ race and ethnicity is summarized in Table 2. Most of the participants lived in Alberta (81%) or British Columbia (12%). Children received their COVID vaccines at vaccination centres (80%), pharmacies (11%), or pop-up vaccination clinics (9%). The majority of children did not have any physical (92%) or mental health (99%) conditions.

Standard care and intervention groups did not differ on sociodemographic variables (Table 1, *p*s > 0.05). Parents assigned to intervention group 1 (i.e., receiving intervention information only) reported higher levels of pain catastrophizing about their child’s pain (*M* = 19.29, *SD* = 11.85) compared to parents randomized to intervention group 2 (i.e., receiving intervention information and verbal instructions), *p* = 0.01. No other statistically significant differences between the groups were detected.

On average, the first vaccination appointment occurred 11 days (*SD* = 16.88) after the consent and receipt of study materials (i.e., pain management strategies, intervention instructions). The memory interview was completed 8.2 days (*SD* = 1.28) after the first vaccination appointment. The second vaccination appointment took place 9 weeks (*SD* = 4.05) after the first appointment. There were no significant differences in children’s reports on pain, pain-related fear, or memory of pain/fear as a function of group allocation, *p* > 0.05.

### 3.1. Primary Outcomes Analyses

#### 3.1.1. Memory of Pain and Pain-Related Fear

Controlling for the initial ratings of pain and fear, there were no statistically significant differences across the groups in children’s recall of their pain intensity (*η_p_*^2^ = 0.05, *p* = 0.16) and pain-related fear (*η_p_*^2^ = 0.01, *p* = 0.83) associated with their first COVID vaccine injection as a function of group allocation (Table 3).

#### 3.1.2. Pain and Pain-Related Fear during the Second COVID Vaccine Injection

Similarly, children did not significantly differ in their reports of pain intensity (*η_p_*^2^ = 0.01, *p* = 0.73) and pain-related fear (*η_p_*^2^ = 0.02, *p* = 0.56) associated with their second vaccination injection as a function of group allocation (Table 3).

#### 3.1.3. Post hoc Power Analyses

Post hoc power analysis with an alpha level at 0.05 indicated a power of 0.62 for the primary outcome of memory of pain, and power of 0.45 for the primary outcomes of memory of fear and future pain and pain-related fear.

### 3.2. Secondary Outcomes Analyses

#### 3.2.1. Intervention Feasibility, Acceptability, and Parental Adherence

Potential participants were recruited in the order they contacted the study team. Twenty-nine interested participants could not be reached. Out of remaining 119 parent–child dyads, 77 were enrolled in the study, resulting in an overall enrollment/recruitment rate of 65%. Ninety-six percent (*n* = 74) of enrolled participants received allocated intervention with 92% of participants (*n* = 71) completing the memory interview and 86% (*n* = 66) completing the study. Parents randomized to either of the intervention groups were motivated to learn the intervention (*M* = 8.91/10, *SD* = 1.44) and understood the intervention principles (*M* = 9.14/10, *SD* = 1.21); the levels of parental motivation and understanding of the intervention did not differ across the two intervention groups, *p*s > 0.05. Parents who received verbal instructions felt connected to the interventionist (*M* = 7.85/10, *SD* = 1.63) with no significant differences across interventionists. Parents rated the intervention as acceptable (*M* = 37.65/45, *SD* = 4.91). The acceptability ratings did not differ across the two intervention groups, *p* > 0.05.

Regardless of group allocation, parents reported using pain management strategies in a similar way for both vaccination injections, *p*s > 0.05. Most parents discussed the upcoming vaccination (81% before the first and second appointments), avoided reassuring their child during or after the vaccine injection (53% before/during the first appointment; 62% before/during the second appointment), and distracted their child (66% during the first appointment; 62% during the second appointment). Fewer parents used numbing creams (34% for the first appointment; 31% for the second appointment) and relaxation techniques (34% before/during the first appointment; 33% before/during the second appointment). Nearly half of all participants created a comfort plan for their child before their first (48%) and second (46%) vaccination appointments.

Parents who received memory-reframing intervention strategies with or without verbal instructions reported using the strategies similarly, *p*s > 0.05. After the first vaccination appointment, most parents talked about positive things that happened before, during, or after the vaccine injection (82%), avoided using pain-related words (57%), and enhanced their children’s pain-related self-efficacy (i.e., highlighted child’s bravery 78%; discussed coping strategies 59%). Most parents did not report noticing any overly distressing memories of pain (57%).

#### 3.2.2. Expected Pain and Pain-Related Fear

There were no statistically significant differences in children’s expected levels of pain (*η_p_*^2^ = 0.04, *p* = 0.26) or pain-related fear (*η_p_*^2^ = 0.01, *p* = 0.81) as a function of group allocation.

#### 3.2.3. Parental and Children’s Pain-Related Self-Efficacy

Parental pain-related self-efficacy significantly increased from baseline (*M* = 6.69/10, *SD* = 2.48) to post-first vaccine injection (*M* = 8.63/10, *SD* = 1.50; *η_p_*^2^ = 0.44, *p* ≤ 0.001); however, there were no differences in the rate of increase as a function of group allocation (*η_p_*^2^ = 0.02, *p* = 0.57). Similarly, children’s pain-related self-efficacy significantly increased from baseline (*M* = 4.66/10, *SD* = 3.33) to post-first vaccine injection (*M* = 7.70/10, *SD* = 3.54; *η_p_*^2^ = 0.31, *p* ≤ 0.001) with no differences as a function of group allocation (*η_p_*^2^ = 0.05, *p* = 0.22).

### 3.3. Exploratory Analyses

#### Individual Predictors of Intervention Efficacy

Data from parent–child dyads allocated to either of the intervention groups were used for these analyses. Controlling for children’s initial pain and fear ratings for the first vaccine injection, parents’ and children’s sociodemographic characteristics, anxiety, vaccine hesitancy, and parents’ tendency to catastrophize about their children’s pain were not significantly correlated with children’s memories of pain and fear or children’s pain and fear ratings for the second vaccine injection, *p*s > 0.005. Similarly, parental ratings of intervention acceptability and feasibility were not associated with primary outcomes. However, among parents who received verbal instructions (i.e., intervention group 2), higher ratings of parent–interventionist rapport were associated with more accurate or more positively biased children’s memories of fear during the first vaccine injection while controlling for the initial ratings of fear associated with the first vaccine injection, *r* = −0.70, *p* = 0.001.

## 4. Discussion

This is the first trial of a parent-led memory-reframing intervention in the context of pediatric COVID vaccine injections, which is important given two doses and repeated boosters that are recommended for continued protection [47,48]. The study tested the efficacy of the intervention to change children’s memories of pain and pain-related fear associated with COVID vaccine injections to be more accurate and/or positively biased (i.e., recall same or lower levels of pain and/or fear compared to the initial ratings; primary outcome). The study also examined the potential of the intervention to decrease the levels of pain intensity and pain-related fear during subsequent COVID vaccine injections (primary outcome). Additionally, the feasibility and acceptability of different modes of intervention delivery (i.e., information only; information and verbal instructions) were examined (secondary outcome). Finally, we examined the effects of the intervention on children’s anticipatory levels of pain intensity and pain-related fear, as well as children’s and parents’ pain-related self-efficacy (secondary outcome). A series of exploratory analyses investigated the effects of individual characteristics and intervention acceptability on the intervention efficacy.

The majority of our hypotheses were not supported likely due to small effect sizes associated with the intervention. Specifically, the intervention did not result in large differences in children’s memories of pain or pain-related fear associated with the first COVID vaccine across groups. Children also rated their expected and actual pain intensity and pain-related fear associated with the second COVID vaccine injection in a similar way across all groups. These findings are in contrast to the previous trial of the parent-led memory-reframing intervention that was efficacious in changing children’s memories of post-surgical pain to be less distressing [22]. Similarly, previous memory-reframing interventions delivered by researchers resulted in more accurate or positively biased memories of pain in the context of procedural and needle pain [16]. However, previous trials of memory-reframing interventions also did not affect anticipatory or recalled pain-related fear associated with dental injections [18], surgery [22], or lumbar punctures [17]. It is possible that their memories of pain-related fear are more salient and/or resistant to change. Similar to anxiety, fear and memory of fear may be more amenable to change via exposure and targeting avoidance behaviours [49].

The small sample size is a notable limitation to consider when interpreting the findings. The demonstrated effect size (i.e., *η_p_*^2^ = 0.05) is comparable to the previous trial of the intervention [22]. Future trials would need to recruit larger and more diverse samples (i.e., at least 100 participants per group) to further examine the intervention effects. Another reason for the lack of significant findings may be related to the study design. Specifically, the previous intervention trials involved an observed session of parent–child reminiscing about a past painful event to ensure that parents engaged in reminiscing about past events involving pain [22,24]. We chose not to observe parent–child reminiscing and, instead, to use parent report of intervention adherence to increase the feasibility of the current project. Most of the parents assigned to the intervention groups did report using the intervention strategies; however, it is unclear how precise their reports were and whether the intervention produced a statistically and clinically significant change in parental reminiscing style (e.g., using fewer pain-related words, discussing emotions associated with the vaccine injection, focusing on helpful coping strategies). Importantly, it is not clear how frequently—if at all—parents reminisced about the first vaccine injection with their children. Completing assigned homework (in this case, homework equated to reminiscing with a child about the first COVID vaccine injection) fully and frequently plays a key role in treatment outcomes [50]. Given many parents’ tendency to avoid or resist discussing past painful experiences [35], external motivation may be needed to ensure that parents engage in the central aspect of the intervention, that is, talking about past events involving needle pain. Future trials should include observed instances of parent–child interactions about past events involving vaccine injections as a way to ensure the reminiscing takes place.

Another reason for nonsignificant findings may be related to the type of past painful event. The past trials of the intervention focused on children’s memories of past surgeries [22,24]. Past surgery may be a more salient and memorable event associated with much higher levels of pain that parents and children discuss more frequently, as compared to a routine needle procedure (i.e., vaccine injection). Indeed, parents of young children, who had undergone a tonsillectomy reported that they discussed the surgery with their children more frequently as compared to another past event that involved physical pain (e.g., a needle procedure) [51]. In the same study, when discussing past surgery, as compared to other events involving pain, parents used more emotion-laden (including positive emotion-related words) and fewer pain-related words [51]. These reminiscing elements align closely with the intervention strategies (i.e., focusing on the positive aspects of the experience, reducing the frequency of pain-related words) [22]. Future iterations of the intervention will need to counteract these naturally occurring reminiscing style elements associated with discussing past nonsurgical pain by highlighting the benefits of emotion-laden language and the drawbacks of pain-related words.

To further increase the feasibility of the study, the timing of the intervention delivery was changed. Specifically, in the current trial, parents received intervention instructions before the first COVID vaccination appointment, as opposed to previous trials that delivered the intervention several weeks following the targeted painful event [22,24]. Despite text reminders that contained key intervention principles and were sent to parents immediately after the first vaccination appointment, parents may have forgotten the previously reviewed intervention strategies and/or were not as motivated to use them. Additionally, if parents did use the intervention strategies upon receiving the prompt immediately after the first vaccination appointment, reminiscing may have interfered with children’s immediate experience of the vaccine injection, as opposed to targeting their memories of the experience. Further, if parents used the intervention principles immediately after the vaccine injection (e.g., immediately shifted to the positive aspects of the experience; focused on coping and bravery), it may have resulted in children feeling confused, distressed, and invalidated. Future trials are needed to determine the best timing of the intervention.

In line with our hypothesis, enrollment/recruitment rate (i.e., 65%) was comparable to previously published studies [22,24]; parent report also indicated high levels of feasibility [38]. Parents randomized to the intervention groups rated the intervention as highly acceptable. The feasibility and levels of acceptability of intervention did not differ across two modes of delivery (i.e., information only versus information and verbal instructions). Indeed, the intervention principles are straightforward and are easily summarized into reading materials and a short video clip. Verbal instructions have not provided any additional boost to parents’ motivation to learn, or their understanding of, the intervention principles. Nor did the verbal instructions significantly change parents’ ratings of how much they liked the intervention or believed that the intervention principles will result in improved outcomes (i.e., intervention acceptability). The intervention (delivered with verbal instructions) was previously rated as highly feasible and acceptable [22,24]. Equal levels of feasibility and acceptability, as well as self-reported use of intervention principles, across two modes of intervention delivery are promising as the intervention may not require additional time and costs needed for the interventionists. However, given the small sample size and sample homogeneity (i.e., socioeconomic status, education level), future larger trials with more diverse samples are needed to replicate the finding. It is possible that in more diverse samples, verbal instructions may be needed to clarify the intervention principles and/or increase caregivers’ motivation to learn and apply the intervention.

Parent and children’s pain-related self-efficacy increased from baseline to post-first vaccine injection regardless of the group allocation. Indeed, accomplishing a task (in this case, receiving the first vaccine injection or supporting a child in this process) increases one’s belief that they are capable of performing this task [52]. Yet, receiving the intervention did not result in higher levels of self-efficacy, which is surprising as the intervention directly targets social persuasion (i.e., receiving assertion from other people that one can accomplish a task) [52]. This finding suggests the need for future potential changes to the intervention (e.g., more pronounced validation and assertion of parents’ and children’s ability to manage pain). These changes should be made with input from patient partners.

A series of exploratory analyses did not reveal any significant associations between parent and children sociodemographic or baseline (i.e., anxiety, needle fear, vaccine hesitancy) characteristics and the effects of the intervention, which is in line with previous findings [22,24]. However, when interpreting these preliminary findings, it is important to consider the study sample. Given that participants sought out the study on their own (as opposed to being recruited by a research team), it is probable that the sample was skewed to include more educated parents with a higher socioeconomic status, lower vaccine hesitancy, and lower needle fears. Intervention trials in more diverse samples may reveal sociodemographic and/or individual characteristics that influence the efficacy of the intervention. Of note, higher levels of parent–interventionist rapport were significantly correlated with a more accurate or more positively biased children’s memories of fear associated with the first vaccine injection. The finding, albeit limited by a very small sample size (i.e., parents who received intervention instructions), is in line with established associations between therapeutic rapport and beneficial therapeutic outcomes [53]. Future trials could employ the potential of therapeutic rapport to enhance intervention effects.

Given the small study sample and small effects of the intervention, any clinical implications associated with the results would be premature. If future trials of the intervention demonstrate larger effect sizes associated with the intervention and continue to find no added benefit of the verbal instructions, larger-scale pragmatic trials would be warranted. The urgency of addressing children’s distress, pain, and fear associated with needle procedures remains. The COVID-19 pandemic is ongoing, and the continued need to obtain vaccination boosters is likely. Any additional ways to alleviate children’s vaccine injection pain and fear need to be investigated further and shared with the public.

Despite the small sample size, it is important to note the study’s strengths. In contrast to some of the previous memory-reframing strategies delivered by researchers in the context of vaccine injection pain [23], this intervention did not deceive children and did not deny their past distress associated with needles. Instead, parents were instructed to detect, acknowledge, and reframe in a validating manner any overly distressing pain memories. Additionally, the study provided evidence-based pain management strategies to all participants. Thus, the lack of significant findings may suggest that memory-reframing strategies do not add any incremental value to children’s memories of, and future experience of, pain and fear; this hypothesis needs to be examined in future trials. Finally, publishing non-significant findings is needed to counter publication bias and ethical principles of research [54].

The trial’s findings should be viewed in light of further limitations. First, the study was underpowered due to the small sample size. Future larger trials of the intervention for children undergoing repeated needle procedures are warranted. Further, the study sample primarily consisted of children aged 4 and 5 years. Children aged 4 years and older are capable of forming pain memories, yet may remember higher levels of pain and fear [55]. Thus, the intervention may have to be adjusted to be effective with younger children. It is also unclear whether the measures used in the study (i.e., FPS-R and CFS) can reliably capture children’s memories of pain and fear. More research is needed to examine the psychometric qualities of these measures in the context of children’s pain memories. Further, future trials should recruit a wider age of youth to examine the intervention efficacy in older and/or younger children. Second, the assessment of pain and pain-related fear associated with the second COVID vaccination injection (i.e., two out of four primary outcomes) was administered by participating parents who were aware of the group allocation, which increases the risk of bias. Outcome assessors who are blind to intervention received (or lack thereof) should assess all primary outcomes in future trials. Third, only interested parents reached out to be enrolled in the study, which might have resulted in a sample of highly motivated participants who may not be representative of the general population. More active recruitment strategies (e.g., recruiting participants at pharmacies and community clinics) would ensure a sample that includes caregivers with varying levels of motivation and a more realistic intervention strategies’ consumption. Fourth, the video produced by the Meg Foundation and included in the intervention materials is publicly available. It is possible that parents assigned to the standard care group watched the video, which could have interfered with the study’s results. Finally, parental adherence to the intervention strategies was self-reported and did not include reports of frequency or duration of the intervention use. Ecological momentary assessments of parent–child conversations may be a potential method of assessing parental adherence while maintaining future studies’ feasibility.

## 5. Conclusions

The present study was a small RCT that examined the efficacy of the parent-led memory-reframing intervention to change children’s memories of vaccine injection pain intensity and pain-related fear, as well as the intervention potential to reduce future pain intensity and fear. The intervention demonstrated small-scale group differences in both the memory of, and future experiences of, pain intensity and pain-related fear. Yet, both methods of intervention delivery (i.e., information only or information and verbal instructions) were equally feasible and acceptable. Given the ongoing pandemic and the continuing need to obtain additional vaccine boosters, future larger trials of the intervention in the context of vaccine injections are warranted.

## Figures and Tables

**Figure 1 children-10-01099-f001:**
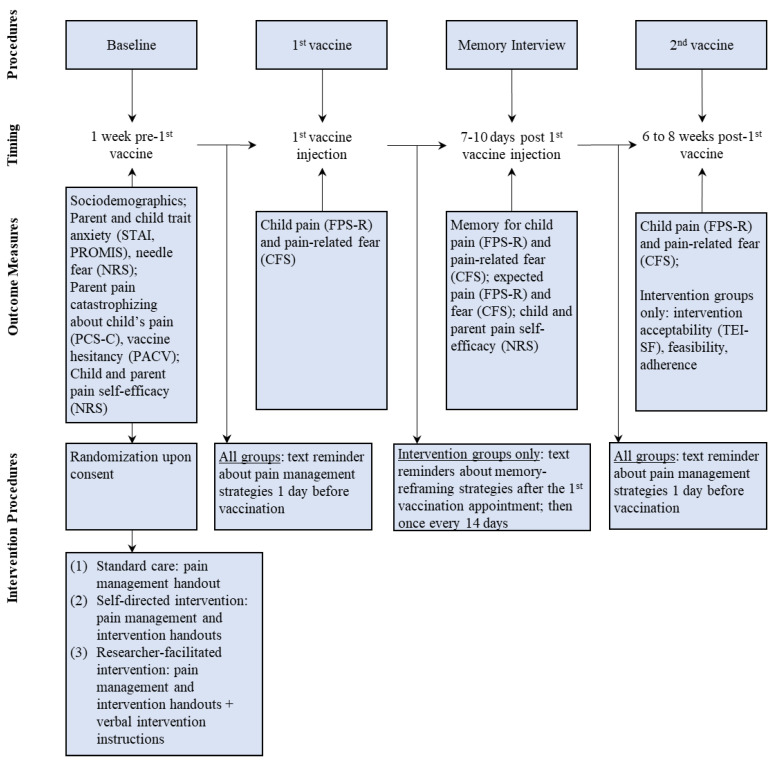
Study procedures.

**Figure 2 children-10-01099-f002:**
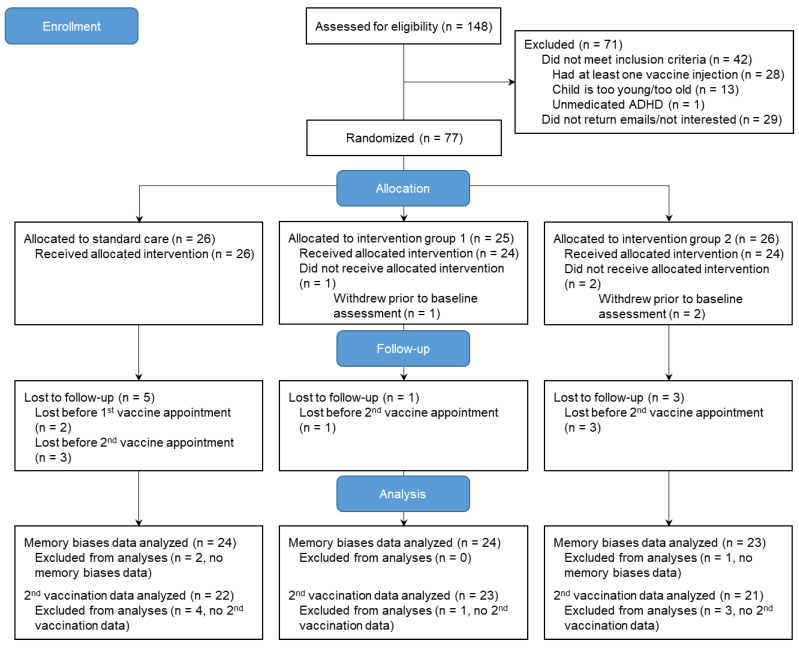
CONSORT flow diagram of trial participants.

**Table 1 children-10-01099-t001:** Demographics and baseline variables by group.

Variable	Standard Care Group (*n* = 26)	Intervention Group 1 (*n* = 24)	Intervention Group 2 (*n* = 24)	Total (*n* = 74)	*p* Value ^a^
Child age, years, *M* (*SD*)	4.75 (1.63)	4.44 (0.50)	4.25 (0.44)	4.49 (1.05)	0.23
Child gender, *n* (%) ^b^					
Female	15 (58)	12 (50)	15 (63)	42 (57)	0.68
Male	11 (42)	12 (50)	9 (38)	32 (43)	
Parent gender, *n* (%)					
Female	23 (89)	21 (88)	22 (92)	66 (89)	0.89
Male	3 (12)	3 (13)	2 (8)	8 (11)	
Child ethnicity, *n* (%)					
Persons of color	13 (50)	7 (29)	6 (25)	26 (35)	0.14
White	13 (50)	17 (71)	18 (75)	48 (65)	
Parent ethnicity, *n* (%)					
People of color	7 (27)	7 (29)	4 (17)	18 (24)	0.56
White	19 (73)	17 (71)	20 (83)	56 (76)	
Annual household income, *n* (%)					
<CAD 70,000	3 (12)	4 (17)	2 (9)	9 (13)	0.75
≥CAD 70,000	22 (88)	20 (83)	21 (91)	63 (88)	
Parent education, *n* (%)					
High school/vocational school/some college	4 (15)	3 (13)	1 (4)	8 (11)	0.52
College/graduate degree	22 (85)	21 (87)	23 (96)	66 (89)	
Marital Status, *n* (%)					
Married/common-law	24 (92)	22 (92)	23 (96)	69 (93)	0.82
Single/divorced	2 (8)	2 (8)	1 (4)	5 (7)	
Parent anxiety (trait), STAI-T, *M* (*SD*)	41.23 (9.23)	41.17 (11.18)	36.50 (8.56)	39.68 (9.83)	0.16
Parent catastrophizing, PCS-P, *M* (*SD*)	15.62 (7.29)	19.29 (11.85)	11.96 (5.74)	15.62 (9.04)	0.02
Parent needle fears, *M* (*SD*)	2.35 (2.45)	2.04 (2.49)	2.50 (3.02)	2.30 (2.63)	0.83
Parent vaccine hesitancy, PACV, *M* (*SD*)	21.72 (20.02)	14.71 (17.17)	15.86 (23.02)	17.57 (20.13)	0.42
Parent pain-related self-efficacy, baseline, *M* (*SD*)	7.23 (2.41)	6.13 (2.19)	6.63 (2.66)	6.68 (2.44)	0.28
Parent pain-related self-efficacy, after first vaccination appointment, *M* (*SD*)	9.17 (1.13)	8.46 (1.47)	8.26 (1.74)	8.63 (1.50)	0.09
Children’s past experience with vaccinations, *n* (%)					
Had at least one vaccination	21 (81)	20 (80)	19 (73)	60 (78)	0.83
Children’s pain intensity associated with past vaccinations, NRS, *M* (*SD*)	5.19 (2.42)	5.65 (2.08)	4.95 (1.93)	5.27 (2.15)	0.59
Children’s pain-related fear associated with past vaccinations, NRS, *M* (*SD*)	6.78 (2.66)	6.24 (2.76)	6.90 (2.53)	7.26 (2.73)	0.47
Child anxiety, PROMIS, *M* (*SD*)	51.19 (7.72)	52.86 (6.75)	50.58 (8.39)	51.54 (7.61)	0.57
Child needle fears, NRS, *M* (*SD*)	6.19 (2.50)	6.92 (2.28)	6.75 (2.25)	6.61 (2.34)	0.52
Child pain-related self-efficacy, baseline, *M* (*SD*)	5.71 (3.17)	3.76 (3.18)	4.36 (3.32)	4.66 (3.28)	0.12
Child pain-related self-efficacy, after first vaccination appointment, *M* (*SD*)	7.96 (3.46)	8.42 (3.16)	6.96 (3.97)	7.79 (3.54)	0.36

Abbreviations: STAI-T, state-trait anxiety inventory (trait); PCS-P, pain-catastrophizing scale-parent version; PACV, parent attitudes about childhood vaccines. ^a^ One-way analyses of variance (ANOVAs), chi-squared tests, or Fisher’s exact tests. ^b^ The percentages may not add up to 100 due to rounding up/down.

**Table 2 children-10-01099-t002:** Race and ethnicity by group.

Variable	Standard Care Group (*n* = 26)	Intervention Group 1 (*n* = 24)	Intervention Group 2 (*n* = 24)	Total (*n* = 74)
Parent ethnicity, *n* (%)				
Arab/West Asian	-	1 (4)	-	1 (1)
Black	1 (4)	-	-	1 (1)
East Asian	4 (15)	3 (13)	1 (4)	8 (11)
Latinx	1 (4)	-	1 (4)	2 (3)
Multiethnic/multiracial	-	1 (4)	-	1 (1)
Other	-	1 (4)	-	1 (1)
South East Asian	1 (4)	1 (4)	2 (8)	4 (5)
White	19 (73)	17 (70)	20 (83)	56 (76)
Child ethnicity, *n* (%)				
Arab/West Asian	-	1 (4)	-	1 (1)
Black	1 (4)	-	-	1 (1)
East Asian	3 (12)	2 (8)	1 (4)	6 (8)
Latinx	1 (4)	-	1 (4)	2 (3)
Multiethnic/multiracial	8 (31)	2 (8)	2 (8)	12 (16)
Other	-	1 (4)	-	1 (1)
South East Asian	-	1 (4)	2 (8)	3 (4)
White	13 (50)	17 (70)	18 (75)	48 (65)

**Table 3 children-10-01099-t003:** Children’s memory biases and second vaccine injection pain intensity and pain-related fear as a function of group.

Criterion	Group (Adjusted Mean, 95% CI) ^a^			Mean Differences (95% CI)			F Value ^b^	*p* Value	η_p_^2^
	Standard Care Group (*n* = 26)	Intervention Group 1 (*n* = 24)	Intervention Group 2 (*n* = 24)	Standard Care vs. Intervention Group 1	Standard Care vs. Intervention Group 2	Intervention Group 1 vs. Intervention Group 2			
Memory of pain intensity (first vaccine injection) (*n* = 71)	2.95 (1.75 to 4.14)	1.59 (0.39 to 2.79)	3.10 (1.87 to 4.33)	1.36 (−0.73 to 3.44)	−0.15 (−2.27 to 1.97)	−1.51 (−3.62 to 0.61)	1.90	0.16	0.05
Memory of pain-related fear (first vaccine injection) (*n* = 71)	1.31 (0.86 to 1.75)	1.20 (0.75 to 1.64)	1.39 (0.93 to 1.85)	0.11 (−0.67 to 0.89)	−0.09 (−0.88 to 0.70)	−0.20 (−0.98 to 0.59)	0.19	0.83	0.01
Pain intensity (second vaccine injection) (*n* = 65)	4.10 (2.49 to 5.70)	3.60 (2.00 to 5.20)	4.51 (2.87 to 6.15)	0.49 (−2.30 to 3.29)	−0.41 (−3.24 to 2.42)	−0.90 (−3.72 to 1.92)	0.31	0.73	0.01
Pain-related fear (second vaccine injection) (*n* = 66)	1.97 (1.30 to 2.65)	1.47 (0.81 to 2.13)	1.80 (1.11 to 2.49)	0.50 (−0.66 to 1.66)	0.17 (−1.02 to 1.36)	−0.33 (−1.51 to 0.84)	0.59	0.56	0.02

^a^ The adjusted means are derived from the analyses of co-variance controlling for corresponding initial pain/pain-related fear ratings. ^b^ F values for the analyses of co-variance.

## Data Availability

The data presented in this study are available upon request from the corresponding author. The data are not publicly available as the data may contain information that could compromise the privacy of research participants.

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
