# Peer review of "A Randomized Controlled Trial of a Parent-Led Memory-Reframing Intervention to Reduce Distress and Pain Associated with Vaccine Injections in Young Children"

_children, 2023, doi:10.3390/children10071099_

Round 1

Reviewer 1 Report

Title:

Page 1, Line 1-3: The inclusion of "COVID" in the title implies that the study focuses on distress and pain specifically related to COVID vaccine injections. However, since the manuscript aims to alleviate distress and pain associated with vaccine injections in general, it might be more suitable to omit the specific reference to COVID from the title. By doing this, the findings and interventions of this study could have broader applicability across different vaccine contexts, extending beyond the scope of COVID.

 Introduction:

Page 1, Line 42-44: It would be beneficial to provide about the prevalence of pain associated with medical procedures, such as vaccine injections, and the factors contributing to it.

Page 2, Line 46-47: The statement that “Children remember their past painful experiences, and those memories predict future levels of pain better than the initial pain experience “would benefit from further elaboration on the underlying psychological mechanisms supporting this finding. Providing a more in-depth explanation of the reasons behind this phenomenon would enhance the clarity and comprehensibility of the statement.

 Page 2, Line 60-63: Authors should provide a succinct overview of the review's conclusions rather than simply directing readers to read the entire review. This will help readers quickly grasp the study's main findings regarding Memory-reframing interventions.

Page 2, Line 72: The authors should provide more details on the intervention itself, including the specific techniques or strategies used, to enhance the understanding of how it may influence the outcomes

Page 2, Line 94: Use of the term "positively-biased memories" requires clarification.

 Page 2, Line 94-95: It would greatly benefit the readers if the terms 'positively-biased/accurate memories for pain and fear' and 'Negatively-biased pain memories' are explained in the context of the manuscript in a more comprehensive and accessible manner.

 2. Materials and Methods:

Page 3, Line 11-112: Organizing the "Materials and Methods" section with proper subheadings is indeed a good practice to enhance readability and understanding for readers. By dividing the section into logical subheadings, authors can provide a clear structure and make it easier for readers to locate specific information within the section.

Page 3, 145-146: The reliability and validity of the FPS-R (Faces Pain Scale-Revised) and CFS (Children's Fear Scale) were examined for assessing pain intensity and pain-related fear in children aged 4 to 12 years. However, a question arises regarding the reliability and validity of these scales in capturing children's memory of pain and fear after a few days following the actual experience. It is important to provide evidence supporting these concerns.

Page 6, Line 230: Authors should include details of verbal instructions given to parents in the manuscript for better clarity, particularly for specific questions and remarks provided by parents assigned to the intervention group.

 Page 6, Line 243: Authors should furnish details of duration and frequency of interventions in the manuscript for more clarity which has been given by parents allocated to the intervention group.

 Page 6, line 237: The pamphlet, in addition to its informative content, also contained a link to a video that effectively elucidated the principles of the intervention. It is worth noting that this video was made readily accessible to the public for the purpose of patient education. Were precautions taken by the authors to prevent any potential contamination in the standard care group?

 Statistical Methods

 Page 9, Line 379: On page 9, line 379, the authors stated that they utilized an Intent-to-Treat (ITT) methodology to analyze the primary outcomes in both datasets. However, it became apparent upon closer examination that the actual analysis deviated from this intended approach. Kindly provide further clarification on this discrepancy.

Page 9, Line 378-379: Page 9, lines 378-379 states the inclusion of "controlling for children's initial pain and fear ratings." Does this refer to the ratings obtained during the administration of the first COVID vaccine or during the memory interview session?

 Results

Page 11, Line 432: In the demographics and baseline variables section, authors are advised to provide comprehensive information regarding previous vaccine injections and any past instances of discomfort experienced by children. This data should be specifically outlined on page 11, line 432(Table 1)

 Discussion

 Page 14, Line 522: Please provide the literature support and references for the statement made on page 14, line 522 regarding the necessity of repeated boosters for continued protection.

 Page 14 Line 523: Please provide clarification on whether the study assessed the efficacy or effectiveness of the intervention.

Title:

Page 1, Line 1-3: The inclusion of "COVID" in the title implies that the study focuses on distress and pain specifically related to COVID vaccine injections. However, since the manuscript aims to alleviate distress and pain associated with vaccine injections in general, it might be more suitable to omit the specific reference to COVID from the title. By doing this, the findings and interventions of this study could have broader applicability across different vaccine contexts, extending beyond the scope of COVID.

 Introduction:

Page 1, Line 42-44: It would be beneficial to provide about the prevalence of pain associated with medical procedures, such as vaccine injections, and the factors contributing to it.

Page 2, Line 46-47: The statement that “Children remember their past painful experiences, and those memories predict future levels of pain better than the initial pain experience “would benefit from further elaboration on the underlying psychological mechanisms supporting this finding. Providing a more in-depth explanation of the reasons behind this phenomenon would enhance the clarity and comprehensibility of the statement.

 Page 2, Line 60-63: Authors should provide a succinct overview of the review's conclusions rather than simply directing readers to read the entire review. This will help readers quickly grasp the study's main findings regarding Memory-reframing interventions.

Page 2, Line 72: The authors should provide more details on the intervention itself, including the specific techniques or strategies used, to enhance the understanding of how it may influence the outcomes

Page 2, Line 94: Use of the term "positively-biased memories" requires clarification.

 Page 2, Line 94-95: It would greatly benefit the readers if the terms 'positively-biased/accurate memories for pain and fear' and 'Negatively-biased pain memories' are explained in the context of the manuscript in a more comprehensive and accessible manner.

 2. Materials and Methods:

Page 3, Line 11-112: Organizing the "Materials and Methods" section with proper subheadings is indeed a good practice to enhance readability and understanding for readers. By dividing the section into logical subheadings, authors can provide a clear structure and make it easier for readers to locate specific information within the section.

Page 3, 145-146: The reliability and validity of the FPS-R (Faces Pain Scale-Revised) and CFS (Children's Fear Scale) were examined for assessing pain intensity and pain-related fear in children aged 4 to 12 years. However, a question arises regarding the reliability and validity of these scales in capturing children's memory of pain and fear after a few days following the actual experience. It is important to provide evidence supporting these concerns.

Page 6, Line 230: Authors should include details of verbal instructions given to parents in the manuscript for better clarity, particularly for specific questions and remarks provided by parents assigned to the intervention group.

 Page 6, Line 243: Authors should furnish details of duration and frequency of interventions in the manuscript for more clarity which has been given by parents allocated to the intervention group.

 Page 6, line 237: The pamphlet, in addition to its informative content, also contained a link to a video that effectively elucidated the principles of the intervention. It is worth noting that this video was made readily accessible to the public for the purpose of patient education. Were precautions taken by the authors to prevent any potential contamination in the standard care group?

 Statistical Methods

 Page 9, Line 379: On page 9, line 379, the authors stated that they utilized an Intent-to-Treat (ITT) methodology to analyze the primary outcomes in both datasets. However, it became apparent upon closer examination that the actual analysis deviated from this intended approach. Kindly provide further clarification on this discrepancy.

Page 9, Line 378-379: Page 9, lines 378-379 states the inclusion of "controlling for children's initial pain and fear ratings." Does this refer to the ratings obtained during the administration of the first COVID vaccine or during the memory interview session?

 Results

Page 11, Line 432: In the demographics and baseline variables section, authors are advised to provide comprehensive information regarding previous vaccine injections and any past instances of discomfort experienced by children. This data should be specifically outlined on page 11, line 432(Table 1)

 Discussion

 Page 14, Line 522: Please provide the literature support and references for the statement made on page 14, line 522 regarding the necessity of repeated boosters for continued protection.

 Page 14 Line 523: Please provide clarification on whether the study assessed the efficacy or effectiveness of the intervention.

Author Response

June 12, 2022

Dear Dr. Vervoort and Reviewers,

Please find enclosed the revision of our manuscript, “A Randomized Controlled Trial of a Parent-Led Memory-Reframing Intervention to Reduce Distress and Pain Associated with COVID Vaccine Injections in Young Children” (children-2444624), which we are submitting for further consideration for publication in the Children’s Special Issue on Psychological Interventions for Pediatric Pain.

We thank you and the reviewers for thorough and constructive feedback. We have carefully considered all of the comments and revised the manuscript accordingly. We think that these revisions have improved the quality and clarity of our manuscript.

Below, we provide point-by-point responses to the reviewers’ comments including our answers to the reviewers’/editor’s questions, rationale for our decisions regarding the changes, and/or a brief explanation of all changes made. Specific changes are referenced in terms of their location (pages and lines) in the marked copy of the manuscript. All changes in the manuscript document have been tracked.

Reviewer 2 Report

Dear colleagues!

Questions of the study of pain and, especially, childhood - perhaps the most difficult.

Noteworthy is the excessive citation, or rather self-citation by the authors of their works - from 46 sources 17 (namely 2, 4, 7, 8, 9, 10, 13, 17, 18, 19, 21, 23, 25, 30, 32 , 33, 46). I hope that this is really important for this work, but please clarify.

On line 121 of the "Materials and Methods" section, you write that "study was advertised on social media (i.e., Twitter, Facebook)". Is this sample of data sources sufficiently reliable?

Did you use any objective methods for assessing pain, or did you just rely on questionnaire data? If not, why not?

Please write the null hypothesis more clearly.

In table 1 in "Results" it is better to replace "People of color" with "person of color"

I think the sample size needs to be recalculated and the criteria used checked to see if there really is no statistically significant difference in the results. On the other hand, it is precisely objective research methods that would help you get more correct results.

Author Response

(The authors gave the same response as above.)

Round 2

Reviewer 2 Report

Hello. The answers are detailed and accurate.

I have no more comments.

Author Response

We thank the Reviewer for their feedback.